# TASK TOKENS: A FLEXIBLE APPROACH TO ADAPTING BEHAVIOR FOUNDATION MODELS

**Ron Vainshtein**
Technion
sronv@campus.technion.ac.il

**Zohar Rimon**
Technion
zohar.rimon@campus.technion.ac.il

**Shie Mannor**
Technion
shie@ee.technion.ac.il

**Chen Tessler**
NVIDIA Research
ctessler@nvidia.com

## ABSTRACT

Recent advancements in imitation learning for robotic control have led to transformer-based behavior foundation models (BFMs) that enable multi-modal, human-like control for humanoid agents. These models generate solutions when conditioned on high-level goals or prompts, for example, walking to a coordinate when conditioned on the position of the robot's pelvis. While excelling at zero-shot generation of robust behaviors, BFMs often require meticulous prompt engineering for specific tasks, potentially yielding suboptimal results. In this work, we introduce "Task Tokens" - a method to effectively tailor BFMs to specific tasks while preserving their flexibility. Our approach integrates naturally within the transformer architecture of BFMs. Task Tokens trains a task-specific encoder (tokenizer), with the original BFM remaining untouched. Our method reduces trainable parameters per task by up to $\times 125$ and converges up to $\times 6$ faster compared to standard baselines. In addition, by keeping the original BFM unchanged, Task Tokens enables utilizing the pre-existing encoders. This allows incorporating user-defined priors, balancing reward design and prompt engineering. We demonstrate Task Tokens' efficacy across various tasks, including out-of-distribution scenarios, and show their compatibility with other prompting modalities. Our results suggest that Task Tokens offer a promising approach for adapting BFMs to specific control tasks while retaining their generalization capabilities.

Recent advances in imitation learning have facilitated the emergence of behavior foundation models (BFMs) designed for humanoid control (Peng et al., 2022; Won et al., 2022; Luo et al., 2024a; Tessler et al., 2024). These models, generate diverse behaviors when trained on large-scale human demonstration data. In this work, we focus on a specific type of BFM, which we call Goal-Conditioned Behavior Foundation Models (GC-BFMs). Methods such as Masked Trajectory Models and MaskedMimic fall into this category (Wu et al., 2023; Tessler et al., 2024). These methods use transformer architectures that process sequences of tokenized goals — high-level objectives such as "follow a path" or "reach with your right hand towards the object" are mapped to embedding tokens. These tokens condition the model's behavior generation. Specifically, we focus on MaskedMimic, which has manifested as a particularly effective framework, demonstrating robust zero-shot generalization (ability to handle new, unseen tasks without additional training) through its token-based goal conditioning mechanism.

For real-world usage, BFMs must be *flexible* enough to solve a variety of tasks, but at the same time *specialized* enough to effectively solve complex tasks. Despite MaskedMimic's proficiency in generating diverse motions from high-level goals, significant challenges persist in defining precise goal specifications, or prompts, for complex tasks. Typically, an environment-specific reward can be designed, but this is prone to potential errors in complex, long-horizon tasks. In contrast, GC-BFMs provide a "prompt-engineering" interface, where the user can specify high-level goals, which can result in a more stable motion, but might be less intuitive for some tasks. Consider a game character tasked with walking to an object and striking it. Even in this simple task, on the one hand, a common emerging error of using reward design is that the character walks backward to the goal, but on the other, specifying high-level goals for the striking motion to precisely hit the target is hard. This

creates a fundamental gap between the model's ability to generate robust and natural motions and the precise control needed for specialized tasks. A unified, flexible, and scalable paradigm for adapting BFMs for many complex downstream tasks, while retaining the original motion robustness, is needed.

To this end, we propose Task Tokens, a novel approach that integrates goal-based control with reward-driven optimization within GC-BFMs like MaskedMimic. Our method, illustrated in Figure 1, establishes a hybrid control paradigm: users provide high-level behavioral priors via goals (e.g., "walk toward the object while facing forward"). These goals are encoded using the pre-existing GC-BFM. Concurrently, the system autonomously learns, via reinforcement learning, task-specific embeddings to optimize dense rewards (e.g., "strike the target with maximum impact"). In this setting, Task Tokens serve to refine and enhance the user-defined goals: they build upon the priors by incorporating reward feedback, allowing the model to achieve more precise and effective behaviors than either approach could alone. This integration leverages the inherent tokenization framework of GC-BFMs, enabling a seamless combination of user-defined and learned conditioning tokens.

Our training paradigm preserves the pretrained BFM's extensive behavioral knowledge. During training, the system generates behaviors from the BFM, conditioned on both user-defined goals and the emergent Task Tokens, optimizing the encoder to produce tokens that align behaviors with task-specific rewards. This strategy ensures that the resulting motions remain consistent with the motion manifold defined by the frozen BFM, ensuring robustness and multi-modality capabilities. Inspired by parameter-efficient adaptation techniques in NLP (Hu et al., 2021; Houlsby et al., 2019; Li and Liang, 2021) our method modifies the model's behavior through a lightweight, trainable module that leverages gradients from the frozen BFM. This allows the Task Token encoder to guide behavior without fine-tuning the full model, making it scalable for solving many downstream tasks.

Our experimental evaluation demonstrates that Task Tokens effectively balance MaskedMimic's ability to generate robust, human-like motions with the precision required for task-specific control. This hybrid framework achieves rapid convergence and high success rates, surpassing traditional hierarchical reinforcement learning methods in sample efficiency and requiring fewer learned parameters. Moreover, by adhering to the BFM's underlying motion manifold, Task Tokens conserve multi-modal prompting capabilities and exhibit stronger generalization across diverse environmental conditions, including variations in friction and gravity. These results demonstrate the potential of our approach to unify goal-based and reward-driven control, enhancing behavior optimization for complex tasks.

CONTRIBUTIONS

- **Task-Specific Adaptation:** We propose Task Tokens, a novel and parameter-efficient approach to adapt MaskedMimic, a Goal-Conditioned Behavior Foundation Model (GC-BFM), to specific tasks via tokenized control, without fine-tuning the foundation model, preserving its zero shot capabilities.

- **Scalability:** Our approach is parameter-efficient, requiring up to $\times 125$ less parameters and converges up to $\times 6$ faster than alternative methods.

- **Hybrid Control Paradigm:** Our method enables a seamless combination of user-defined high-level priors (e.g., textual or joint-based goals) with learned reward-driven optimization.

- **Performance and Generalization:** Task Tokens match the high task performance of full fine-tuning, while surpassing other methods in terms of robustness to changes in environment dynamics, such as gravity and friction.

## 1 RELATED WORK

**Humanoid Control:** Humanoid control is a challenging domain spanning both robotics and computer animation, with the shared goal of generating realistic and robust behaviors. In the animation community, physics simulation is used to ensure the generated motions are realistic and enable the characters to react to dynamic changes in the environment. To achieve this, they typically leverage imitation learning methods combined with motion capture data to learn and generate human-like behaviors in new and unseen scenarios (Peng et al., 2018; 2021; Luo et al., 2023; Tessler et al., 2023; Gao et al., 2025). Similar approaches are observed in the robotics community, with the addition of sim-to-real adaptation used to ensure the controller can overcome the imperfect modeling of the world by the simulator (Viceconte et al., 2022; Lu et al., 2024; Ji et al., 2024).

Our work builds on these foundations by preserving their flexibility and robustness while enabling precise task-specific control.

**Behavior Foundation Models:** Recent advances in reinforcement learning have led to the development of Behavior Foundation Models (BFMs) that can generate diverse behaviors for embodied agents. PSM (Agarwal et al., 2024) and FB representations Touati and Ollivier (2021) provide a framework for learning policies conditioned on a target stationary distribution. These models perform remarkably well when the requested behavior can be represented by a stationary distribution (for example, using a reward-weighted combination of data samples). However, covering the entire space of solutions in high dimensional control tasks remains a challenge for these models. Methods such as Adversarial Skill Embeddings (Peng et al., 2022, ASE) and PULSE (Luo et al., 2024a) overcome this limitation by constraining the policy to reproducing human demonstrations. First, they compress a large repertoire of human reference motions into a latent generative policy. Then using reinforcement learning they train a hierarchical controller (Sutton et al., 1999) to pick the latent (motion to perform) at each step and solve new and unseen tasks.

In this work, we focus on Goal Conditioned Behavior Foundation Models (Chen et al., 2021; Zitkovich et al., 2023; Tessler et al., 2024). In contrast to the aforementioned methods, GC-BFMs can solve new and unseen tasks without specific training by directly mapping from goals to actions. Their mode of operation can be seen as a form of inpainting, where the model attempts to reproduce the most likely outcome given the training data for any provided objective. However, this strength is also a limitation, as these models struggle when presented with out-of-distribution constraints, such as those defined manually by a user or task. Our Task Tokens approach addresses this limitation by providing a mechanism to incorporate task-specific optimization while preserving the model's ability to generate natural, human-like behaviors.

## 2 Preliminaries

Our proposed method leverages MaskedMimic to effectively solve a specific distribution of humanoid tasks by learning a "task encoder" with reinforcement learning.

### 2.1 Reinforcement Learning

A Markov Decision Process (Puterman, 2014, MDP) models sequential decision making as a tuple $M = (\mathcal{S}, \mathcal{A}, P, R, \gamma)$. At each time step $t$ the agent observes a state $s_t \in \mathcal{S}$ and predicts an action $a_t \in \mathcal{A}$. As a result, the environment transitions to a new state $s_{t+1}$ based on the transition kernel $P$ and the agent is provided a reward $r_t \sim R(s_t, a_t)$. The objective is to learn a policy $\pi : \mathcal{S} \to \mathcal{A}$ that maximizes the expected discounted cumulative reward $\pi^* = \mathrm{argmax}_{\pi \in \Pi} \mathbb{E}\pi \left[ \sum_t \gamma^t r_t \right]$.

### 2.2 MaskedMimic

MaskedMimic presents a unified framework for humanoid control, extending goal-conditioned reinforcement learning through imitation learning. Goal-Conditioned Reinforcement Learning (GCRL) involves augmenting the state space with a goal $g$, allowing a policy $\pi(s|s, g)$ to map states and desired goals to appropriate actions, effectively enabling a single policy to solve multiple tasks. Unlike traditional GCRL approaches that learn from reward signals, MaskedMimic learns directly from demonstration data through online distillation (Ross et al., 2011, DAgger). By combining a transformer architecture with random masking on future goals represented as input tokens, MaskedMimic learns to reproduce human-like behaviors from various modalities, such as future joint positions, textual commands, and objects for interaction. When trained on vast amounts of human motion capture data, this goal-conditioned approach allows MaskedMimic to generalize to new objectives without additional training, all while preserving the similarity to the training data. This combination of architecture and control scheme makes it an ideal foundation for our Task Tokens method, which further enhances its capabilities by learning task-specific tokens to optimize for downstream tasks.

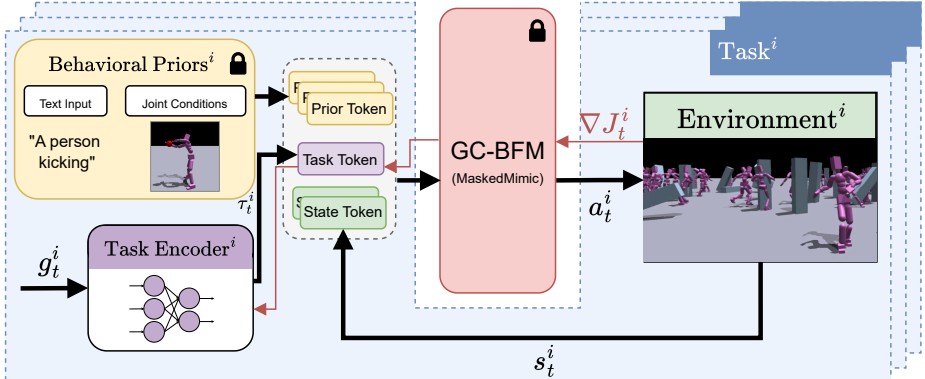

Figure 1: **Task Tokens:** Our approach combines three input sources: (1) Prior Tokens: optional tokens enabling user-defined behavioral priors from text prompts or joint conditions, (2) Task Token: generated by our learned Task Encoder that processes the current goal observation $g_t^i$, and (3) State Token: representing the current environment state $s_t^i$. The prior and state tokens are generated using the pre-trained encoders from the GC-BFM model. The frozen GC-BFM integrates these inputs to produce natural, task-optimized actions $a_t^i$. During training, the policy gradient objective is computed with respect to the BFM's actions, with gradients flowing through the frozen GC-BFM to the Task Encoder, enabling task-specific optimization without modifying the foundation model's parameters.

## 3 METHOD

BFMs excel at producing a wide range of motions, but optimizing them for specific tasks presents significant challenges. Downstream applications often require specialized behaviors that fall outside the common distribution of motions. Traditional approaches to achieve such behaviors involve either time-consuming "prompt engineering" (Tessler et al., 2024) or fine-tuning procedures that risk compromising the rich prior knowledge encoded in the BFM.

### 3.1 TASK TOKENS

The transformer-based architecture of MaskedMimic provides a natural mechanism for integrating new task-specific information. By design, transformers process sequences of tokens, allowing for flexible input composition. Task Tokens can be applicable to any transformer-based BFM capable of attending over arbitrary token sequences. The only requirement is that the model can integrate additional tokens into its input, allowing the Task Encoder to optimize task-specific signals while the BFM itself remains frozen. This enables us to seamlessly incorporate additional tokens without modifying the core network structure.

Our method, Task Tokens (Figure 1), leverages the tokenized nature of the BFMs' objectives. We propose to train a dedicated task encoder to produce specialized token representations for each new task. This task token encapsulates the unique requirements and constraints of the target behavior, providing a concise yet informative signal that can guide the foundation model toward task-specific outputs while preserving its general behavioral priors.

### 3.2 TASK ENCODER

The Task Encoder processes observations that define the current task goal $g_t^i$, represented in the agent's egocentric reference frame and predicts a Task Token $\tau_t^i \in \mathbb{R}^{512}$. These observations vary by task – for instance, in a steering task, they include the target direction of movement $\in \mathbb{R}^2$, facing direction $\in \mathbb{R}^2$, and desired speed $\in \mathbb{R}$, resulting in $g_t^i \in \mathbb{R}^5$. As MaskedMimic is trained to reach future-pose goals, the task encoder is also provided with proprioceptive information. This aligns the encoder with the pre-trained representations, ensuring it can provide meaningful target objectives (see ablation studies in Section E).

We implement the task encoder as a feed-forward neural network. Its output—the Task Token—is concatenated with tokens from other encoders in the BFM's input space. This creates a token "sentence" where the task encoder's outputs represent specialized "words" that guide the model towards achieving the specific task while maintaining natural motion. We provide additional details on the architectural structure of the encoder in Section A.

### 3.3 TRAINING

To optimize the Task Encoder for new downstream tasks, we use Proximal Policy Optimization (Schulman et al., 2017, PPO). During training, the BFM predicts action probabilities based on the combined input tokens (including the learned task token). We compute the PPO objective with respect to the task-specific reward and the BFM's action probabilities. This approach ensures the BFM provides meaningful gradients for updating the task encoder parameters, while the BFM itself remains frozen. This design choice is fundamental – while fine-tuning the entire model might yield higher task-specific returns, it would compromise the BFM's prior knowledge, resulting in less natural and robust motions.

Leveraging MaskedMimic's token-based architecture, Task Tokens require only ~200K parameters per task—compared to ~20M for conventional methods—making them a highly parameter-efficient solution.

## 4 RESULTS

We evaluate the effectiveness of our Task Tokens approach through a comprehensive set of experiments. We examine four critical aspects of our method to validate its performance and applicability. First, we assess the capability of Task Tokens to effectively adapt MaskedMimic for various downstream applications, demonstrating significant improvements in task-specific performance and efficiency (Section 4.1) over the original zero-shot method. Second, we analyze whether the resulting controller preserves the robustness characteristics inherent to the original Behavioral Frequency Modulation (BFM) framework, confirming that stability under variable conditions remains consistent (Section 4.2). Third, we investigate the natural and human-like quality of the generated motions through a human study (Section 4.3). Finally, we explore the synergy of Task Tokens and other prompting modalities, combining effects that further demonstrate the versatility of our method (Section 4.4). These experiments collectively demonstrate that our approach successfully balances task-specific adaptation with the preservation of desirable properties from the foundation model, while requiring significantly less parameters than other baselines.

We provide accompanying video visualizations for all experiments in `sites.google.com/view/task-tokens`, and the code to reproduce all results can be found in the supplementary material.

**Tasks:** We evaluate our approach on a diverse set of humanoid control tasks, all simulated in Isaac Gym (Makoviychuk et al., 2021). For all experiments, we simulate the SMPL humanoid Loper et al. (2015) which consists of 69 degrees of freedom. We focus on the following tasks: **Reach**, the agent must reach a randomly placed goal with its right hand; **Direction**, the agent must walk in a randomly chosen direction; **Steering**, this task combines walking and orienting toward (look-at) random directions; **Strike**, here the agent must reach and strike a target placed at a random location; and **Long Jump**, based on the SMPL-Olympics benchmark (Luo et al., 2024b), the objective is to run and jump as far as possible from a target location. Sample images are shown in Figure 2, full technical details can be found in Section B.

**Baselines and Evaluation** We compare our Task Tokens approach against several competitive baselines: **Pure RL**, a policy trained directly using PPO without leveraging any foundation model; **MaskedMimic Fine-Tune**, using the reward signal to optimize all of the MaskedMimic model without freezing; **MaskedMimic (J.C. only)**, the original MaskedMimic model using only joint conditioning (J.C.) as the prompting mechanism. We use the J.C. defined in the original MaskedMimic for the Reach, Direction, and Steering tasks. In addition, we compare against two state-of-the-art humanoid control baselines: **PULSE**, a hierarchical approach that re-uses a latent space of skills from motion capture data; and **AMP** Peng et al. (2021), which uses a discriminator to ensure motion

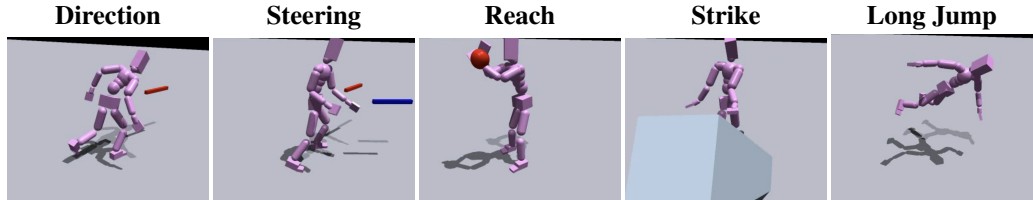

Figure 2: **Multi-task adaptation.** Task Tokens is an effective approach to adapt BFMs to new downstream tasks, while preserving its prior knowledge. Task Tokens can be used alongside other prompting modalities to generate personalized and robust motions to solve new tasks.

Table 1: **Downstream tasks adaptation.** We compare the success rates on the various tasks. While the reward provides a proxy for the policy to learn, the success metric measures the actual task outcome. For example, in Strike this is whether the target object is knocked down. We report the mean and standard deviation of the success rate across 5 random training seeds, except J.C. only which uses zero-shot MaskedMimic and thus reports no variance.

| Method | Reach | Direction | Steering | Long Jump | Strike |
|---|---|---|---|---|---|
| Task Tokens (ours) | **94.88 ± 1.99** | **99.26 ± 0.79** | **88.69 ± 4.04** | **99.75 ± 0.57** | 76.61 ± 3.49 |
| MaskedMimic (J.C. only) | 24.77 | 2.19 | 3.83 | - | - |
| MaskedMimic Fine-Tune | **93.70 ± 4.59** | **99.10 ± 1.29** | **87.44 ± 6.79** | 47.36 ± 54.78 | **83.07 ± 5.71** |
| PULSE | 83.96 ± 2.20 | 97.60 ± 0.62 | 40.72 ± 7.64 | **99.37 ± 1.40** | **83.18 ± 2.67** |
| AMP | 57.14 ± 4.80 | 5.14 ± 0.68 | 4.28 ± 1.42 | 76.59 ± 43.42 | 52.21 ± 47.58 |
| PPO | 89.90 ± 3.25 | 97.74 ± 1.40 | 32.64 ± 40.21 | 61.91 ± 52.26 | **81.36 ± 1.41** |

quality while optimizing for task performance. **Task Tokens** is used with joint conditioning on the relative tasks. Long Jump and Strike pose a great challenge in this sense, thus J.C. is not available for them neither in Task Tokens nor in MaskedMimic.

For all experiments, we report the mean and standard deviation of the success rate across 5 random seeds, representing the variance across independently trained models. Trend lines in the figures indicate the mean performance, with shaded regions denoting one standard deviation across these 5 random seeds. An exception is the J.C. Only model, for which no variance is reported as it represents the performance of a single, re-used MaskedMimic base model evaluated without task-specific retraining. Success rate definitions for each task are listed in Section B, and technical details on the training and evaluation setups are available in Section C.

## 4.1 Task Adaptation

We first show that we can use Task Tokens to effectively adapt MaskedMimic to downstream tasks. For each downstream task, we train a unique task encoder. In the Reach, Direction, and Steering environments, we also use the joint conditioning presented in MaskedMimic (we test the effect of this choice in Section E). Visualizations of the resulting motions can be seen in Figure 2.

We present the numerical results in Table 1. The results show that Task Tokens obtains a high score across the majority of environments, with PULSE, MaskedMimic Fine-Tune, and PureRL obtaining higher scores on the Strike task. Moreover, in Figure 3, we present the evaluated success rate during training. Here, we observe that Task Tokens converges within approximately 50 million steps, while PULSE reaches the same performance around 300 million steps. To achieve these results, Task Tokens requires training an encoder with ~200k parameters, whereas PULSE and MaskedMimic Fine-Tune require 9.3M and 25M parameters, higher by factors of ×46.5 and ×125 respectively. This efficiency is critical in real-world settings where training large models is expensive. Our approach scales to many tasks with minimal additional overhead. These results show that Task Tokens can *effectively* and *efficiently* be used to adapt BFMs, like MaskedMimic, to new unseen tasks.

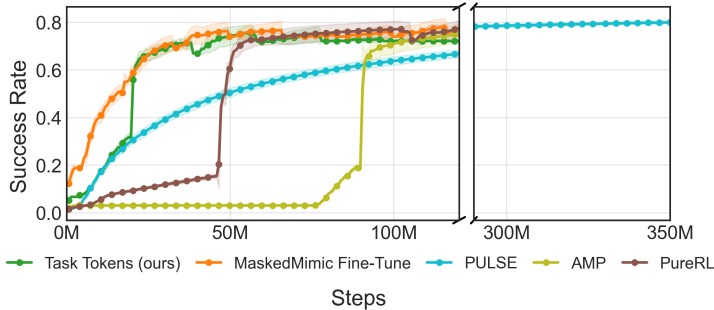

Figure 3: **Convergence for Strike.** Task Tokens is sample efficient, adapting to new tasks in under 50M steps. Fine-tuning converges similarly, likely due to its higher capacity, leading to overfitting.

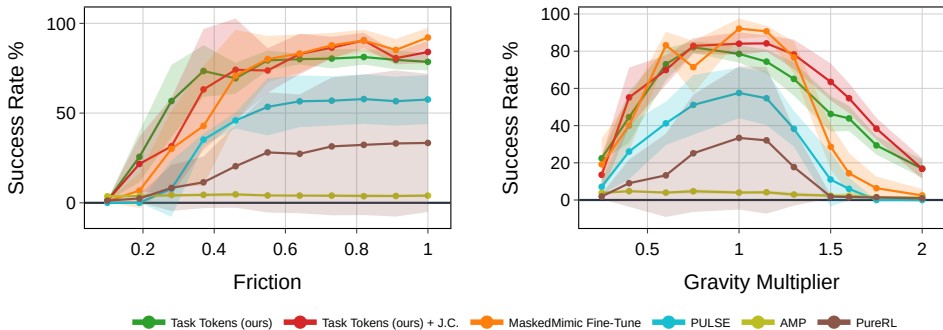

Figure 4: **Out-of-distribution perturbations.** We test the success rate on the steering task when changing the ground friction (on the left) and gravity (on the right). Task tokens (both with and without J.C.) exhibit improved robustness.

## 4.2 OOD GENERALIZATION

The premise of using MaskedMimic is that it has been pre-trained on vast amounts of data and scenarios, which in turn should result in more robust behavior to new and unseen tasks. To test this, we compare on out-of-distribution (OOD) perturbations, not seen during training both in the original BFM and in Task Tokens. We consider changes in both gravity and ground friction.

Indeed, the results, Figure 4, show that by utilizing a BFM, Task Tokens demonstrate highly improved robustness to the new and unseen scenarios. First, it performs almost as well as fully fine-tuning MaskedMimic on the baseline task (no perturbations), outperforming all other baselines. Then, it performs significantly better compared to the baselines as the perturbation increases. Notably, Task Tokens exhibit a *significantly higher success rate* in very low friction scenarios (e.g., $\times 0.4$) and very large gravity (e.g. $\times 1.5$).

As can be seen in the gravity perturbations, fine-tuning the model seems to harm its built-in robustness, leading to worse performance at higher gravity when compared to our minimal intervention approach.

## 4.3 HUMAN STUDY

In some scenarios, such as animation, it is of interest to adapt and generate new behaviors (solutions to tasks) while preserving motion quality. We evaluate the realism of the generated motions by performing a preliminary human study. In our study, we presented ~100 anonymous participants with video triplets. The participants were required to choose the motion that looked more human-looking. We provide additional details in Section F.

Table 2: **Human study, Task Tokens win rate.** We report the percentage by which Task Tokens was deemed more human-like. Higher values means Task Tokens was deemed more human-like.

| Wins v.s. Algorithm | Direction | Steering | Reach | Strike | Long Jump |
|---|---|---|---|---|---|
| MaskedMimic (J.C. only) | 95% $\pm$ 2% | 75% $\pm$ 6% | 53% $\pm$ 5% | - | - |
| MaskedMimic Fine-Tune | 99% $\pm$ 1% | 90% $\pm$ 4% | 85% $\pm$ 6% | 85% $\pm$ 5% | 94% $\pm$ 2% |
| MaskedMimic F.T. + J.C. | 96% $\pm$ 3% | 89% $\pm$ 5% | 82% $\pm$ 6% | - | - |
| PULSE | 15% $\pm$ 5% | 46% $\pm$ 6% | 36% $\pm$ 9% | 24% $\pm$ 5% | 39% $\pm$ 5% |
| AMP | 92% $\pm$ 3% | 84% $\pm$ 4% | 70% $\pm$ 6% | 68% $\pm$ 6% | 96% $\pm$ 3% |
| PPO | 99% $\pm$ 2% | 93% $\pm$ 4% | 89% $\pm$ 5% | 82% $\pm$ 4% | 94% $\pm$ 3% |

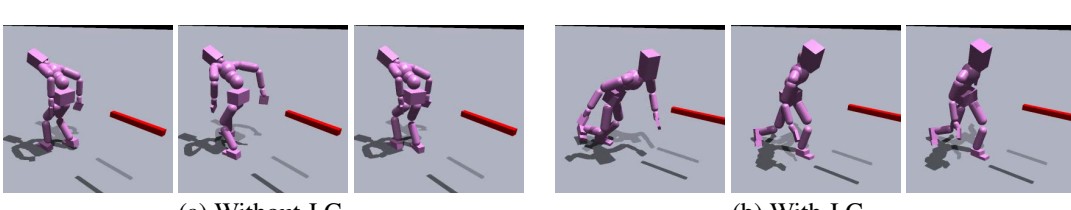

(a) Without J.C.          (b) With J.C.

Figure 5: **Multi-modal prompting.** When trained on the direction task, the policy often learns to walk backwards. Task Tokens enable adding human-defined priors through additional tokens. By combining orientation priors, the BFM is instructed to face the movement direction.

Table 2 describes the percentage of times Task Tokens outperformed each alternative. First, we can see that Task Tokens outperforms MaskedMimic (J.C. only) and MaskedMimic Fine-Tune. This suggests that the user-designed conditions are out-of-distribution for the base MaskedMimic model, and fine-tuning is a less effective way to adapt it compared to Task Tokens in terms of motion quality. In addition, we observe that despite Task Tokens resulting in faster convergence, fewer parameters, and better performance, PULSE scores higher in terms of human-likeness of the motion. It is likely due to the fact that PULSE constrains the high-level representation to stay closer to the prior, whereas MaskedMimic doesn't. Incorporating such a constraint with Task Tokens might improve the motion quality even further. We leave this investigation to future work.

From the results above, we infer that Task Tokens offer a good balance in the tradeoff between efficiency, motion quality, and robustness.

### 4.4 MULTI-MODAL PROMPTING

While some objectives are easy to define through target goals, others are easier to define through rewards. Here, we show how Task Tokens can be trained alongside human-constructed priors (tokens) to achieve more desirable behaviors. We showcase two scenarios, one in the Direction task and another in Strike.

The Direction task provides a reward for moving in the right direction but does not consider the humanoid's orientation. As a result, the policy may converge to walking backwards. While this behavior achieves high reward and success metrics, it is an unwanted behavior. In Figure 5 we show that by combining human-designed priors, providing a target height and orientation for the head, the training converges to an upright forward-moving motion.

An additional challenge is Strike. In this task, the agent needs to hit a target. An emergent behavior is walking backwards toward the target and then performing a "whirlwind" motion, where the agent swirls in circles to hit the target with its hand. In Figure 6 we showcase a combination of 2 prior modalities. First, conditioning on the orientation (similar to the Direction task) the agent is instructed to face the target during the locomotion phase. Then, once close to the target, the agent is guided to strike the target with its foot using a textual objective "a person performs a kick". A more detailed explanation is readily available in the appendix Section D.

Notably, we observed that fine-tuning the entire model leads to the well-known catastrophic forgetting phenomenon (Goodfellow et al., 2015), impairing its ability to retain and integrate such multi-modal

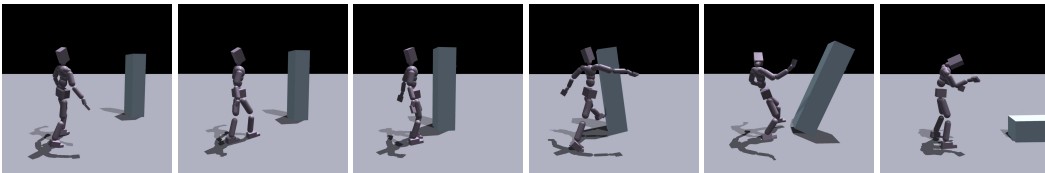

Figure 6: **Incorporating user priors:** Joint and text-based goals can be provided alongside the task-specific task tokens. Here, the user-defined objectives guide the motion style, ensuring the character faces the object while walking upright, and strikes it using a kick.

prompts. In contrast, Task Tokens preserve the pre-trained prompting capabilities by keeping the foundation model frozen, enabling coherent integration of learned and human-specified behaviors. PULSE, in comparison, does not support multi-modal prompts at all, as it lacks the mechanism to condition on diverse inputs beyond goal embeddings.

## 5 LIMITATIONS AND FUTURE WORK

While Task Tokens offer an efficient and effective mechanism for adapting Behavior Foundation Models (BFMs) to downstream tasks, this work also illuminates several promising avenues for future exploration. The performance of Task Tokens is inherently linked to the expressivity and coverage of the underlying pretrained BFM. Future work could thus investigate methods to identify or mitigate gaps in BFM knowledge for target tasks. To enhance the naturalness of generated behaviors, especially for humanoids, future work could explore incorporating methods that better enforce a human-likeness prior within the BFM, such as discriminative methods.

Our current experimental validation has primarily demonstrated the efficacy of Task Tokens with the MaskedMimic architecture. Further research is therefore needed to empirically verify and adapt the methodology across a broader spectrum of GC-BFM architectures, which would solidify the approach's generality. Additionally, while the design of task-specific reward functions and observation spaces currently requires domain expertise, future research could explore methods to (semi-)automate this process, potentially lowering the barrier to entry.

A crucial next step involves investigating the transferability of Task Token-adapted policies to real-world robotic systems, a process that will require addressing inherent sim-to-real challenges. Another key future direction involves extending validation to genuinely robotic tasks beyond simple animation, encompassing complex agentic behaviors that require high-level decision-making in unpredictable environments, thus rigorously testing Task Token efficacy for robust real-world interaction.

The current approach trains a separate, lightweight task encoder for each task. Scaling this to multi-task or lifelong learning scenarios by exploring shared, compositional, or continually learned Task Token encoders presents an interesting challenge. Investigating whether the agent correctly utilizes task information, for instance by examining attention maps associated with the Task Token, could offer valuable insights. such analysis would help verify the mechanism's effectiveness and potentially guide improvements in task encoder design.

Finally, investigating more sophisticated architectures for the Task Encoder itself, beyond the current feed-forward network, could unlock further performance gains. Addressing these areas will further enhance the Task Tokens framework and advance the development of more versatile, adaptable, and capable humanoid agents.

## 6 SUMMARY

This work introduces Task Tokens, a novel approach for enhancing MaskedMimic as a Goal-Conditioned Behavior Foundation Model (GC-BFM). Our method enables a hybrid control paradigm: users provide high-level behavioral priors, which are augmented by task-specific embeddings learned via a Task Encoder trained with reinforcement learning. The encoder maps observations to goal tokens, optimizing dense rewards while keeping the BFM's robustness and multi-modal capabilities.

Experimental results demonstrate that Task Tokens effectively balance efficiency, task precision, and robustness. They achieve rapid convergence and high success rates, surpassing existing methods with superior generalization on OOD cases. This is accomplished while maintaining multi-modal prompting capabilities and exhibiting unmatched parameter efficiency. Human studies further confirm the human-likeness of the generated movements. However, despite these benefits, the approach relies on the quality of the underlying BFM and is currently restricted to simulated environments.

By enabling more flexible and parameter-efficient adaptation of BFMs to specific tasks, this work presents a promising avenue for creating nuanced, responsive, and controllable humanoid agents. This could lead to more realistic character animations and capable humanoid robots capable of tailored behaviors across a wide range of tasks, balancing prompt engineering with reward design.

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

## A  TASK ENCODER ARCHITECTURE DETAILS

The Task Encoder is intentionally designed as a lightweight and generic module. It is implemented as a feed-forward multilayer perceptron (MLP) with ReLU activations, operating on continuous task-goal observations expressed in the agent's egocentric frame. Prior to being processed by the network, all input features are normalized using a running mean–standard deviation estimator computed online during training.

The MLP consists of a small number of fully connected layers and produces a fixed-dimensional output, yielding a Task Token $\tau_t^i \in \mathbb{R}^{512}$. We do not employ residual connections, attention mechanisms, or recurrent components in the Task Encoder. Preliminary experiments indicated that such additions were unnecessary for stable optimization or final performance in this setting, and that a simple architecture sufficed given the expressiveness of the underlying behavior foundation model.

This design choice emphasizes that task adaptation in our framework is primarily mediated through the token interface with the frozen BFM, rather than through architectural complexity or task-specific engineering within the encoder itself.

# B    Environments Technical Details

The controllers operate at 30 Hz, and the simulation runs at 120 Hz.

The tasks are designed to test the versatility and adaptability of the models across a range of real-world scenarios, each adding layers of complexity to the control problem.

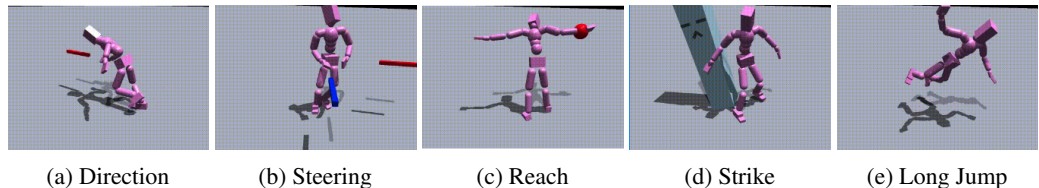

(a) Direction          (b) Steering          (c) Reach          (d) Strike          (e) Long Jump

Figure 7: Visual samples from the tasks.

**Direction**: This task involves directing the character to move in a specific direction. We test the model's ability to control basic locomotion and alignment with a target direction. We measure success by the humanoid's speed in the target direction not deviating from the target speed by more than 20% in the measurement period.

**Steering**: This task requires the humanoid to move in a specific direction while also ensuring that it faces some orientation with its pelvis. This tests more nuanced proceeding motion of the model, creating more diverse scenarios. Success is defined by the character not deviating from the target direction speed by more than 20% while also not deviating from the direction of facing by a sum greater than $45°$.

**Reach**: For this scenario, we task the humanoid with reaching a specified coordinate with the right hand. This requires precision of movement to achieve the specific target. The success is measured by reaching a distance within 20cm between the right hand's position and the target position.

**Strike**: Here we challenge the model to make the character walk toward a target and, once within range, perform an action to knock down the target. This task tests the model's ability to handle both locomotion and more intricate, task-oriented behaviors, involving precise timing and spatial awareness. Success is then defined by the target falling to its side in some orientation and not deviating from it by more than ~$78°$.

**Long Jump** The character is tasked with committing a run in a meter-wide corridor, then jumping over a line after 20 meters, not touching the ground after crossing the jump start line. Success is defined by achieving a jumping distance greater than 1.5 meters.

# C    Training and Evaluation Details

Each experiment seed was trained for 4000 epochs on 1024 parallel environments, totaling approximately 120 million frames. This results in roughly 1–2 GPU-days per seed. PULSE was also trained on 120 million frames, but using only 128 parallel environments. All experiments were run on a single NVIDIA A100 or V100 GPU per seed.

In the main Task Tokens results, we used a Task Encoder implemented as an MLP with hidden layers of size $[512, 512, 512]$, and a critic MLP with layers of size $[1024, 1024, 1024]$. We additionally concatenated the current positions of the head and pelvis joints to the Task Encoder input, which led to slightly more human-like motion.

All reported results reflect the mean and standard deviation of the success rate across the 5 random training seeds. In performance plots, the trend line shows the mean, and shaded areas represent one standard deviation across seeds. Note that the J.C. Only baseline does not report variance as it corresponds to the performance of the single, re-used MaskedMimic base model evaluated without task-specific retraining.

Each reported result corresponds to the final model checkpoint after 4000 training epochs. Note that success rates in training and evaluation may differ due to differences in episode termination criteria.

For training, we used the original published hyperparameters of PULSE. All other methods were trained using MaskedMimic's published hyperparameters. Please refer to the respective publications for additional implementation details.

For AMP, we use a locomotion-focused motion dataset (e.g., walking, running, and transitions), chosen to remain stylistically aligned with the original MaskedMimic corpus. While AMP can achieve stronger performance when provided with task-specific motion data tailored to each skill, such curation is not assumed in our setting. This choice reflects our goal of evaluating task adaptation without per-task motion dataset design, thereby emphasizing the benefit of reusing a generic, pretrained model.

## D    MULTI-PHASE PROMPTING VIA A FINITE-STATE MACHINE

For tasks that naturally decompose into sequential phases (e.g., locomotion followed by interaction), we employ multi-phase prompting using multiple prior tokens in conjunction with a single Task Token. The selection of the active prior token is governed by a simple finite-state machine (FSM). This mechanism enables structured, phase-dependent behavior without introducing multiple task-specific encoders.

Concretely, in the **Strike** task, the agent first locomotes toward the target while being conditioned on an orientation prior that encourages facing the target. Once the agent reaches a predefined striking distance, the FSM transitions to a second state, activating a prior token that encodes a kicking motion (specified via a textual objective). This transition is deterministic and based solely on geometric proximity, allowing smooth phase switching during a single rollout.

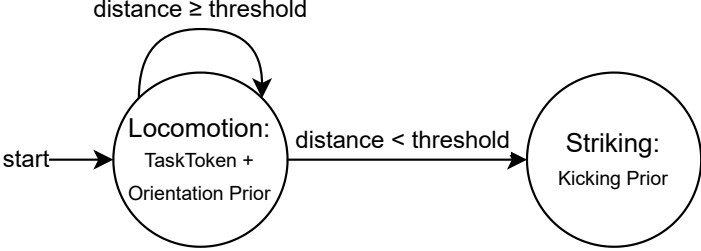

Figure 8: Finite-State Machine for the Strike task. The agent transitions from a locomotion phase to a striking phase based on geometric proximity to the target.

## E    ABLATION STUDY

We experimented with several variables, constructing the Task Encoder. The results are shown in Table 3.

### E.1    MASKEDMIMIC ADAPTATION PARADIGM

Table 3: **Algorithm training scheme ablations**. While fine-tuning the whole MaskedMimic model can produce performing results, we've shown it lacks the human-like abilities of Task Tokens.

| Method | Reach | Direction | Steering | Long Jump | Strike |
|---|---|---|---|---|---|
| Task Tokens (ours) | **95.37 ± 1.80** | 96.89 ± 4.33 | 83.66 ± 5.66 | **99.75 ± 0.57** | 76.61 ± 3.49 |
| Task Tokens (ours) + J.C. | **94.88 ± 1.99** | **99.26 ± 0.79** | 88.69 ± 4.04 | - | - |
| MaskedMimic (J.C. only) | 24.77 | 2.19 | 3.83 | - | - |
| MaskedMimic F.T. | **93.70 ± 4.59** | **99.10 ± 1.29** | 87.44 ± 6.79 | 47.36 ± 54.78 | **83.07 ± 5.71** |
| MaskedMimic F.T. + J.C. | 92.88 ± 3.42 | **98.86 ± 0.32** | **96.41 ± 4.94** | - | - |

Results in Table 3 demonstrate superior performance when using J.C. when available. It is worth noting that F.T. + J.C. means fine-tuning while using the joint priors and does not imply preserving the multi-modal prompting capabilities.

### E.2 TASK ENCODER ARCHITECTURE

We further present some architectural changes made to the Tak Encoder and their effect on output performance, listed in Table 4. Bigger MLP denotes using $[512, 512, 512]$ size MLP encoder versus $[256, 256]$ and Using Current Pose denotes whether the current positions of the head and pelvis are concatenated to the input alongside the task goal. When using joint conditioning, the performance stays high for every choice except when using a smaller encoder with current pose information. This result replicates when not using joint conditioning.

Table 4: **Task Encoder architectural ablations.**

| Method | Bigger MLP | Using Current Pose | Steering Success Rate |
|---|---|---|---|
| Task Tokens (ours) + J.C. | True | True | 87.77 ± 7.14 |
| Task Tokens (ours) + J.C. | True | False | 87.58 ± 7.02 |
| Task Tokens (ours) + J.C. | False | False | 86.88 ± 6.65 |
| Task Tokens (ours) | False | False | 84.28 ± 7.72 |
| Task Tokens (ours) | True | False | 83.30 ± 10.06 |
| Task Tokens (ours) + J.C. | False | True | 79.47 ± 4.71 |
| Task Tokens (ours) | True | True | 78.59 ± 8.93 |
| Task Tokens (ours) | False | True | 66.31 ± 13.11 |

## F HUMAN STUDY TECHNICAL DETAILS

To assess the quality of the motions generated by our method we conducted a human study. The participants were met with this description before filling out the form:

> In this study, you will watch three short videos side by side each time.
>
> These videos show different ways a character moves to complete a task. Your job is to decide which movement looks the most human-like.
>
> Each video is labeled A, B, or C — the labels are shuffled every time. Just pick the one you think does the best job.
>
> Don't worry — there's no right or wrong answer! We just want your opinion.

We used Google Forms to create 3 forms, each containing 40 questions - 8 questions for each of the tasks listed in Section B. In each question, we showed 3 videos side-by-side of 3 randomly sampled sequences generated by the algorithms. We ensured that Task Tokens was presented every time. Joint conditioning was used when applicable, i.e. for Direction, Steering and Reach tasks. The participants were asked to choose which algorithm looks most human-like for the task described in the question. An example

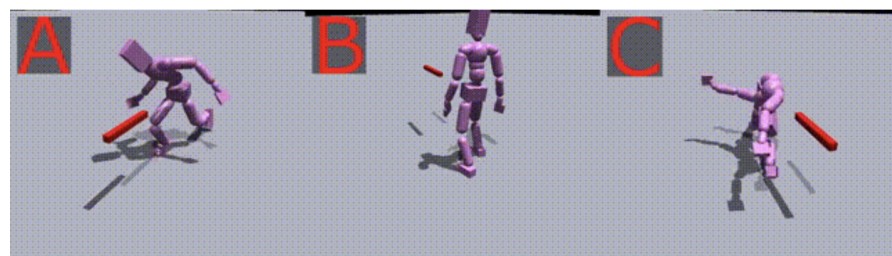

Q 1: Which example looks more human-like for task **walking in red direction**? *

○ A

○ B

○ C

To ensure no bias, we shuffled the order of algorithms in every question and captured the videos from similar angles. The number of participants was 96: 20, 24, 52 for the forms respectively. We analyzed the winning rate for each environment as winning percentage$_{env}^{A}$ = $\frac{\text{\# Task Tokens chosen}}{\text{\#Task Tokens chosen}+\text{\#Algorithm } A \text{ chosen}}$. None of the participants received any compensation for participation.

