# OpenReview forum: "Task Tokens: A Flexible Approach to Adapting Behavior Foundation Models"
_ICLR.cc/2026/Conference — ICLR 2026 Poster_

### Official Review · Reviewer_zBs9 · 2025-10-23

**Soundness:** 3
**Presentation:** 4
**Contribution:** 3
**Rating:** 8
**Confidence:** 3

**Summary:**

This paper introduces Task Tokens, a parameter‑efficient adaptation mechanism for goal‑conditioned behavior foundation models (GC‑BFMs) such as MaskedMimic. A lightweight task encoder maps task observations to a learned token that is appended to the BFM’s input sequence alongside optional human priors; the BFM is kept frozen, and PPO updates only the task encoder via gradients flowing through the frozen model. Across several humanoid control tasks, Task Tokens achieve competitive success rates to full fine‑tuning with far fewer trainable parameters, improved robustness to friction/gravity perturbations, and preserved multi‑modal prompting. A human study indicates motions are generally more human‑like than PPO/AMP and fine‑tuned MaskedMimic.

**Strengths:**

The paper is well‑motivated. It clearly articulates the gap between prompt‑only GC‑BFMs and reward‑driven task optimization, and proposes a neat integration that leverages the transformer token interface without touching the backbone. Freezing the foundation model to avoid collapse/forgetting while steering behavior through a small encoder is a sound and practical design choice to preserve the motion manifold. Empirically, the method is consistently competitive with full fine‑tuning while using ~200k parameters per task, and it outperforms standard RL and AMP/PULSE on several tasks. The study on combining learned tokens with human priors (orientation and text prompts) convincingly demonstrates directability that full fine‑tuning appears to erode. The experimental section is generally decent with five seeds, ablations on the task‑encoder architecture, OOD tests, and a user study. I especially appreciate the explicit discussion of limitations around human‑likeness and reliance on the pretrained BFM compared to PULSE.

**Weaknesses:**

My main issue is the overstatement of some contributions. The method is a logical adoption of ideas from NLP to BFMs, RL training for these tasks is a sensible way to tune the performance of the model, and the results are decent. There is no need to exaggerate other parts of the paper.

Statements on the criticality of using up to “×125 fewer parameters” seem misplaced when the baseline methods train only approximately 25M parameters. This is especially true when training on humanoid simulations, where the sim is also likely a major bottleneck. If the efficiency gains are truly massive, it would be great to have a comparison of GPU‑hours, memory, and inference latency. In addition, it seems like the baseline fine-tuning does not employ standard parameter‑efficient fine‑tuning methods inside the BFM (e.g., LoRA/adapters), which would be the real baseline when talking about efficiency.

The robustness claims are also inflated. “Order of magnitude” gains at very low friction values are only caused by the baseline going slightly faster towards zero. At that point, any small increase in success rates (i.e. going from 0.01% to 1%) is going to be orders of magnitude larger. Given the huge standard deviations, it is even questionable if the result is statistically significant.

Finally, scalability remains unclear: training one encoder per task is simple but offers no multi‑task, compositional, or continual learning path, and the claim that fine‑tuning catastrophically forgets multi‑modal prompting is largely anecdotal without a quantitative retention test on the original prompts.

**Questions:**

- Figure 3 shows several saw-tooth-like drop‑offs for MaskedMimic fine‑tune and Task Token. What causes these instabilities?
- AMP often trails PPO and even fails on Direction/Steering. Why is that when it’s specifically designed for these scenarios? Did you retune AMP per task?
- Efficiency: please report wall‑clock training time (GPU‑hours) and inference throughput for Task Tokens vs. full fine‑tune and PPO/AMP. Does the 6× speedup in steps translate to real time?
- Could you compare against PEFT baselines applied inside the BFM (LoRA, adapters) with comparable parameter budgets? Where does Task Tokens win/lose relative to these?
- Section 4.4 mentions two phases (direction prior then semantics “kick”). Are you using multiple task token adapters here or just multiple prior tokens with a single task token? If multiple task tokens are used, how are they scheduled/combined?
- Token analysis: do task tokens cluster across runs/rollouts and across tasks? Visualizing the resulting token embeddings across training runs for different tasks could reveal whether clustering suggests scalability.

Overall, this is a well‑motivated and seemingly effective adaptation mechanism that preserves the strengths of GC‑BFMs. Addressing the missing PEFT baselines, clarifying efficiency in wall‑clock terms, explaining AMP’s underperformance, and adjusting the framing of some sections would strengthen it further.

---

> ### Author Response · Authors · 2025-11-20
>
> We greatly appreciate your positive feedback: “The paper is well‑motivated…leverages the transformer token interface without touching the backbone” and “Freezing the foundation model to avoid collapse/forgetting while steering behavior through a small encoder is a sound and practical design choice.”
>
> We address the concerns and questions below.
>
> ## Overstatement of parameter efficiency.
>  The advantage of parameter efficiency is primarily in downstream usage rather than raw training speed. In robotics applications, an agent may need to maintain a growing library of skills, with a high-level controller switching between them on demand. Compact Task Encoders make this feasible, allowing multiple skills to reside in memory simultaneously and enabling efficient behavior swapping, which would be challenging with full fine-tuning of large models.
>
> ## Wall-Clock Training Time
> We chose to report training efficiency in terms of environment steps and not wall-clock training time (as is also common in the literature) since many of the chosen baselines are based on separate open-source implementations. The GPU runtime is highly affected by specific technical details and optimizations. Further, we emphasize that TaskToken is not meant to improve inference throughput compared to other adaptation methods (e.g., fine-tuning).
>
> ## PEFT baselines and comparisons.
> "It seems the baseline fine-tuning does not employ standard parameter-efficient fine-tuning methods inside the BFM (e.g., LoRA/adapters)."
> PEFT approaches such as LoRA primarily modify or inject parameters directly into the model weights, making them closer in spirit to fine-tuning. In contrast, TaskTokens take a fundamentally different route: instead of altering the BFM, they generate a task-dependent token that guides the frozen model’s behavior while preserving its pre-trained knowledge and multi-modal capabilities. This non-invasive design avoids the risk of disrupting the BFM’s generality and enables seamless multi-modal conditioning.
>
> While it is possible that incorporating LoRA could strengthen joint-conditioning baselines, exploring PEFT-based weight editing is outside the scope of this work. Our focus is to clearly demonstrate the effectiveness, efficiency, and conceptual simplicity of TaskTokens as a lightweight, behavior-elicitation mechanism. Comparing TaskTokens to PEFT baselines remains a promising direction for future research.
>
>
> ## Inflated robustness claims.
>  Our reported robustness results focus on the meaningful perturbation regimes rather than extreme tails. For instance, at friction ×0.4 and gravity ×1.5, baseline performance drops sharply while TaskTokens maintains reasonable success rates (~70% and ~50%, respectively). We acknowledge that the phrase “orders of magnitude” overstates the effect and will revise it to more accurately reflect these comparisons.
>
> ## Unclear scalability.
>  While a single Task Token handles only one task, training multiple TaskTokens enables multi-task and compositional usage. At inference, different encoders can be efficiently switched based on the current sub-goal. For instance, a long-horizon task can be decomposed by a VLM into sub-goals, with each sub-goal addressed by a dedicated Task Token, allowing compositional and sequential behavior without retraining the underlying BFM.
>
> ## Training instabilities.
>  These saw-tooth drop-offs are also observed in the PureRL baseline (~100M steps) and reflect common RL training dynamics. Temporary dips in performance can occur when an optimization step causes short-term underperformance, which is typically corrected in subsequent updates. Such fluctuations are inherent to the stochastic nature of RL.
>
> ## AMP underperformance.
>  AMP underperforms compared to PPO because, unlike PPO, it optimizes both the task reward and a style reward. This dual objective requires carefully selecting motion data whose style aligns with the task, which is nontrivial. PPO, by contrast, focuses solely on maximizing the task reward, allowing it to achieve higher performance in a potentially unnatural manner.
>
> ## Multi-phase prompting.
>  In this setup, we use multiple prior tokens alongside a single Task Token. The prior tokens are managed via a simple finite-state machine: when the agent reaches striking distance from the object, we switch to the prior token that encodes the kicking motion. This approach allows smooth multi-phase behavior without needing multiple task-specific encoders.
>
> ## Token clustering analysis.
>  We agree that analyzing token embeddings across runs and tasks is a promising direction. Such clustering studies could provide insights into the structure, reusability, and scalability of TaskTokens, and we consider this an inspiring avenue for future research.

---

> > ### Comment · Reviewer_zBs9 · 2025-11-22
> >
> > Thank you for your response to the review.
> >
> > I appreciate the clarification on the motivation for memory requirements. While the reasoning behind not fine-tuning the model to maintain its full behavior distribution and generalizability is clear, LoRAs are standard for fine-tuning, enable the same kind of parameter-efficient switching, and would have thus been an interesting comparison. I acknowledge, however, that the time until the end of the rebuttal is not sufficient for adding this baseline.
> >
> > The order of magnitude improvement still seems to overstate the results, unless I have missed something. I.e., PULSE achieves around 40% success at 0.4x friction, whereas TaskToken is at 70%.
> >
> > For a thorough comparison with AMP, I would have expected that suitable data was collected for the style reward. Otherwise, it is not surprising that it performs so badly.
> >
> > I thank the authors for clarifying the points on scalability, the training stability, and multi-phase prompting. It might be helpful to include a sentence on the latter (specifically the state machine) in the paper.

---

> > > ### Author Response · Authors · 2025-11-26
> > >
> > > Thank you for the thoughtful follow-up.
> > >
> > > ## On the robustness comparison and “order-of-magnitude” wording.
> > > We appreciate the careful reading of the robustness plots. After re-examining the numbers, we agree that _“orders of magnitude”_ overstates the effect across the full perturbation range. Our intention was to emphasize that in the more challenging regimes - particularly near the edges - baselines such as PULSE often drop to very low success rates, while TaskTokens remain in a meaningful performance range. We will adjust the wording to reflect this more accurately.
> > >
> > > ## On the AMP comparison.
> > > Regarding AMP, the motion dataset consisted of locomotion-oriented clips (walking, running, transitions) to stay stylistically aligned with the original MaskedMimic corpus. Naturally, if one curates task-specific motion data for every new skill, AMP can perform significantly better. However, the goal of this experiment was to evaluate adaptation without such per-task data design, highlighting the benefit of reusing a generic model rather than constructing bespoke motion sets for each task. We will clarify this better in the paper.
> > >
> > > ## On clarifying multi-phase prompting.
> > > We will add a clarifying sentence about the simple finite-state machine used to switch between prior tokens during multi-phase tasks.
> > >
> > > Thank you again for the constructive feedback.

---

### Official Review · Reviewer_MgVR · 2025-10-27

**Soundness:** 3
**Presentation:** 2
**Contribution:** 3
**Rating:** 6
**Confidence:** 2

**Summary:**

The paper proposes an approach to adapt pre-trained Behavior Foundation Models (BFMs) to downstream tasks by training "task tokens" -- special tokens which are added to the prompt of BFM, which represent the task-specific information. These tokens are trained via RL from environment rewards. They are modeled as neural networks which map task goals to a fixed tokens dimension.

The authors conduct multiple experiments to demonstrate the efficiency of their approach compared to baselines -- finetuning BFMs directly, pure RL without BFMs and other approaches which make use of motion capture data.

Overall, the proposed approach is simple, concise and efficient.

**Strengths:**

* The approach is sound, makes sense, and simple
* The experimental evidence suggests that it works well in practice, despite being much cheaper than alternatives
* The approach essentially makes use of "prompt engineering" by making it automatic and training "tokens" from data

**Weaknesses:**

* I think the technical presentation of the paper could be improved:

** The "task tokens" is the major contribution of the paper, but it does not spend enough time discussing different design choices regarding "Task encoder" architecture. It feels that authors just tried a few simple things and it "just worked". However, I think the research community will benefit greatly from a proper analysis of the different impacts of design choices in such architecture. I.e., do they use some form of normalization ? What are activation functions? Are there residual connections and if not, why not? I think also trying to conduct a more careful study here will end up in a method which eventually performs even better than what the authors have

** Table 1 looks a bit suspicious to me. The results of the baselines are inconsistent. You see that PPO performs well on Strike, Reach and Direction but achieves significantly lower performance on Steering and Long Jump. Similar can be said about other methods. Why is that? What it looks to me could be that the authors did not spend enough effort trying to make sure that the baselines are optimized carefully.

Minor: Since the proposed method is much cheaper than the baselines, it would be very helpful to have a figure which shows total FLOPs spent compared to baselines.

**Questions:**

* What are the design choices which are important for the task encoder? Can you do a more careful scientific study here so that readers understand what things matter?

* Can you explain discrepancies in Table 1? Why baselines numbers are inconsistent?

---

> ### Author Response · Authors · 2025-11-20
>
> We thank the reviewer for noting that “The approach is sound, makes sense, and simple” and for acknowledging the cost-effectiveness of TaskTokens.
>
> We appreciate the reviewer’s engagement with our work and offer our responses below.
>
> ## Insufficient technical presentation of Task Encoder architecture.
>  We intentionally focused on simple and efficient Task Encoder architectures. Given that our tasks are continuous, we used a straightforward MLP with ReLU activations and applied running mean–std normalization to the input features. We did not include residual connections, as preliminary experiments indicated they were not necessary for stable training in this setting. We will clarify these design choices in the paper to provide a more complete technical description.
>
> ## Inconsistent baseline results in Table 1.
>  All experiments were conducted using a shared set of hyperparameters across methods, drawn from prior work (AMP, ASE, CALM, MaskedMimic, PULSE). We did not fine-tune hyperparameters for individual tasks, which explains the variability in baseline performance across tasks. Notably, TaskTokens achieved competitive results under the same settings, highlighting its ability to adapt effectively to new tasks with minimal task-specific tuning.
>
> ## FLOPs comparison
> While TaskToken provides little computational overhead on top of the base BFM, it is not designed to reduce the overall FLOPs, and the total FLOPs using TaskTokens will obviously increase (as we add an additional encoder). We chose to focus on the parameter efficiency in the paper, as it is crucial for scaling to many task adaptations, while FLOPs are not (as one TaskToken will be used per given task).

---

> > ### Comment · Reviewer_MgVR · 2025-11-24
> > **Response**
> >
> > I would like to thank the authors for their response. I am keeping my current score.

---

### Official Review · Reviewer_up9Q · 2025-11-01

**Soundness:** 3
**Presentation:** 4
**Contribution:** 4
**Rating:** 8
**Confidence:** 4

**Summary:**

This paper presents "Task Tokens," a novel framework for adapting large-scale Behavior Foundation Models (BFMs), such as MaskedMimic, to specific downstream tasks. They propose to use a parameter-efficient approach that trains a new, lightweight "Task Encoder" for each specific task using reinforcement learning (PPO) while freezing the pretrained network to avoid catastrophic forgetting. This small encoder maps current task-specific goals (e.g., target coordinates) to a "Task Token." This token is then concatenated with the BFM's standard inputs (like state tokens and user-defined "prior tokens") to adapt the BFM's behavior

The proposed method allows the PPO gradient to flow through the frozen BFM, updating only the small Task Encoder. This encoder learns to "steer" the general-purpose BFM to solve the specific RL task while keeping the BFM's learned prior of natural human motion. The method is up to 125x more parameter-efficient and converges 6x faster compared to finetuning the whole BFM network. It also preserves the BFM's robustness in out-of-distribution (OOD) scenarios where standard fine-tuning fails.

**Strengths:**

**Elegant and Novel Solution to Important Problem** The main contribution is the application of a light-weight task encoder to steer large-scale behavior models. The method, by design, prevents catastrophic forgetting by freezing the core BFM. This is an elegant and, to my knowledge, novel solution (at least in the field of RL and robot learning) to address the balance of general policy and specific task performance. The effectiveness of such an approach opens up the opportunity to better control the behavior of robot foundation models.

**Solid Empirical Validation** The experiments show the proposed method is highly parameter-efficient, and it also preserves the foundation model's robustness in some cases compared to finetuning (see Figure 4). The human study (Table 2) confirms that the resulting motions are perceived as more human-like than those from baselines like PPO, AMP, or naive fine-tuning, demonstrating that the BFM's prior is indeed preserved. The framework (Sec 4.4, Fig 5/6) also allows for a seamless combination of two control modes: user-defined "Prior Tokens" (like text or joint conditions) provide high-level style guidance, while the learned "Task Token" optimizes for the low-level, reward-specific objective. This hybrid approach is highly flexible and a step forward in usability.

**Weaknesses:**

**Limited Task Complexity:** This is the main weakness. The tasks evaluated (Direction, Steering, Reach, Strike, Long Jump) are primarily reactive motor skills or goal-reaching tasks. While diverse, they do not appear to require complex, long-horizon composition of tasks. It is unclear if the "Task Token" approach, which optimizes a single, continuous token, is sufficient to adapt the BFM to tasks requiring multi-stage sequential logic (e.g., "find the block, pick it up, and place it in the correct box").

**Reward and Human-likeness Trade-off:** The human study (Table 2) reveals that Task Tokens were consistently rated as less human-like than PULSE. The authors suggest PULSE "constrains the high-level representation to stay closer to the prior." This implies that the RL-driven Task Token, in its pursuit of maximizing the reward, may be finding solutions that are on the edge of the BFM's natural motion manifold—less human-like, but more optimal for the task.

**Scalability:** The current approach trains one separate Task Encoder for each new task. The paper's scalability claim (Contribution #2) refers to *per-task* efficiency (a small ~200k encoder vs. a 25M model). It also highly depends on the reward definition to adapt to certain tasks.

**Questions:**

Besides some questions raised in the weakness section, I have some additional questions:

Q1. It's not very clear to me what exactly the inputs to the task encoder are. Are they the same observation as the BFM? Is some goal information concatenated to the task encoder?

Q2. Is the BFM sensitive to the exact position of the task tokens in the input sequence?

Q3. Could there be some task tokens that represent some more "generation" modifications to the model? This reminds me of a paper altering the behavior of VLA models via some "activation steering" [1]. For example, one might train a task encoder that makes the humanoid move faster or move with lower torso height, and these encoders may be useful to alter the model's behavior in different tasks, along with more task-specific encoders.

Reference:
[1] Häon, B., Stocking, K.C., Chuang, I., & Tomlin, C.J. (2025). Mechanistic interpretability for steering vision-language-action models. ArXiv, abs/2509.00328.

---

> ### Author Response · Authors · 2025-11-20
>
> We sincerely appreciate your positive remarks: “Elegant and Novel Solution to Important Problem” and “Solid Empirical Validation”.
>
> We are grateful for the reviewer’s constructive comments and respond to them below.
>
> ## Limited Task Complexity.
>  We see TaskTokens as a modular component within a broader robotics pipeline. Long-horizon tasks are often tackled by first decomposing them into sub-goals using reasoning models (e.g., VLMs). These sub-goals are typically short to medium horizon and largely non-compositional - exactly the setting where TaskTokens excels. By efficiently encoding and reusing such sub-goals, TaskTokens provides a flexible foundation for constructing more complex, multi-stage behaviors.
>
> ## Reward and Human-likeness Trade-off.
>  We agree that RL-driven TaskTokens can produce motions that deviate from the natural motion manifold in pursuit of reward, which may reduce perceived human-likeness compared to PULSE. Incorporating techniques such as trust-region constraints relative to the original policy could help mitigate this effect, keeping the adapted behavior closer to the BFM’s learned distribution while still allowing task-specific optimization.
>
> ## Scalability concerns.
>  Our scalability claim emphasizes per-task efficiency: each Task Encoder is compact (~200k parameters) compared to the full 25M-parameter BFM, making it feasible to train many diverse tasks. Once trained, multiple TaskTokens can be selectively activated at inference according to the task, enabling flexible multi-task deployment without retraining the underlying BFM.
>
> ## Task encoder inputs.
>  The Task Token encoder takes as input task-specific information, which can differ from the observations used by the BFM. For example, in the strike task, while the BFM may be trained on future joint positions and rotations, the Task Token encoder is only provided with a vector specifying the direction and distance to the target. This is described in Section 3.2.
>
> ## Token position sensitivity.
>  This is an important point and depends on the specific BFM architecture. For MaskedMimic, temporal information is already embedded within the input vectors prior to tokenization. Consequently, the model is agnostic to the position of TaskTokens in the input sequence and does not rely on explicit positional encoding.
>
> ## Generative task tokens.
>  Yes, TaskTokens can indeed be used to encode more general behavioral modifications. Analogous to Textual Inversion[1], one can learn tokens that capture constraints or stylistic variations in motion - for example, moving faster or adopting a lower torso posture. These tokens can complement task-specific encoders, allowing flexible control over the model’s behavior. Moreover, the pretrained MaskedMimic already provides capabilities to condition specific joints, such as lowering the torso, which can be combined with TaskTokens as demonstrated in Section 4.4 on multi-modal prompting.
>
> [1] Gal et al. An Image is Worth One Word: Personalizing Text-to-Image Generation using Textual Inversion (2022)

---

> > ### Comment · Reviewer_up9Q · 2025-11-23
> >
> > Thank the authors for the clarification. I'm satisfied with the answers and will maintain the positive rating.

---

### Official Review · Reviewer_Vh4Q · 2025-11-03

**Soundness:** 2
**Presentation:** 3
**Contribution:** 2
**Rating:** 6
**Confidence:** 4

**Summary:**

In this work, the authors propose Task Tokens, where a pre-trained behavior foundation model is then adapted for a new task by using RL to learn a conditioning latent. In this method, the authors first train a behavior foundation model that takes a "goal" through some task encoder as a conditioning. Then, given a new task specified through a reward function, the task token itself is optimized with RL. The authors evaluate this method through task performance and efficiency. In addition, they analyze the stability and the human preference of the results. In these performance analyses, the authors show favorable results for their method.

**Strengths:**

1. The demonstrated performance of the model in the videos is quite impressive.
2. The idea itself is simple and leads to impressive results.
3. The authors evaluate it on a diverse set of tasks and on good set of robustness evaluation.

**Weaknesses:**

1. The method is only evaluated in one setup with one behavior foundation model, MaskedMimic. Evaluation in other settings, such as manipulation, would go a long way to establish the generality of this method.
2. It is hard to evaluate how "new" each of the elicited behaviors are. I.e. how well can this method elicit behaviors that were and were not in the training set of the original training dataset of the BFM? Especially tasks like directions should be in the dataset, and yet the Joint Conditioning method fails to elicit this behavior, which makes it seem the JC method is not great at eliciting pre-trained behavior.
3. The authors make the "human-like" behavior of the prompting method a big part of the study, but devote very little space to understanding the difference between this and PULSE which outperforms task tokens. A further understanding of this difference would be appreciated.
4. In figure 3, the Y axis label is in % but the labels are between (0-1).

**Questions:**

1. What are the major challenges in adapting this method to a manipulation setup?
2. How well can you teach this method tasks that were not in the original dataset? I imagine full fine-tuning can learn any new tasks, vs. without fine-tuning you will not be able to learn something sufficiently off-distribution. But how can we evaluate it?
3. What's the reason behind this method taking equivalent amount of environment steps to reach success as full-fine tuning? My intuition says you should be able to reach comparable performance much faster since this is RL in a much smaller space.

---

> ### Author Response · Authors · 2025-11-20
>
> We thank the reviewer for noting that “The demonstrated performance of the model in the videos is quite impressive” and for appreciating the simplicity and diversity of our evaluations.
>
> We appreciate the reviewer's careful evaluation of our work. Our responses follow.
>
> ## "Evaluation in other settings, such as manipulation, would go a long way to establish the generality of this method."
> We agree that manipulation is an exciting direction for demonstrating the generality of TaskTokens. At the time of submission, however, no manipulation-capable behavior foundation model was available for adaptation. Emerging models such as MaskedManipulator[1] provide exactly this kind of foundation, and we view extending TaskTokens to manipulation tasks as a natural next step for future work.
>
> ## "How well can this method elicit behaviors that were and were not in the training set of the original training dataset"
>  We view BFMs as following a trajectory similar to large generative models in language and vision: although the training corpus often contains the building blocks needed for many tasks, reliably eliciting those behaviors is nontrivial. Joint Conditioning (JC) functions analogously to prompting - sometimes effective, but often sensitive and difficult to control. This is precisely why approaches like Textual Inversion[2] (which motivates TaskTokens) have emerged: they allow optimizing a small set of parameters through the frozen model to more directly access and steer its latent capabilities. TaskTokens serve this role for BFMs, helping reveal both pretrained and novel behaviors that simple conditioning may struggle to extract.
>
> ## Insufficient analysis of PULSE comparison
>  While PULSE often produces more human-like motion, this does not imply that TaskTokens yields unnatural behavior. Instead, our results point to a tradeoff: PULSE achieves strong realism but requires significantly more trainable parameters and computational effort per task. In contrast, TaskTokens offers a far more efficient adaptation mechanism - several orders of magnitude smaller - while still producing coherent behaviors. In addition, PULSE is limited to single-modal prompting, whereas TaskTokens naturally supports multimodal task specifications.
>
> ## Figure 3 presentation error
> Thank you for pointing this out - the axis label is indeed inconsistent. We will correct the figure accordingly in the final version.
>
> ## "What are the major challenges in adapting this method to a manipulation setup?"
>  Two main challenges arise when extending our method to manipulation. First, it requires a BFM trained on an articulated agent with fingers (or another form of grasping capability) along with motion data that includes fine-grained hand interactions. Second, the BFM should ideally encode some understanding of object and scene interactions. Recent models such as MaskedManipulator begin to address these needs, and we see integrating TaskTokens with such BFMs as a natural avenue for future work.
>
> ## "How well can you teach this method tasks that were not in the original dataset?"
> Our working assumption is that large-scale motion datasets such as Motion-X++ (120k+ sequences) provide enough diversity for the BFM to generalize and synthesize solutions for many tasks, even if they are not explicitly represented. TaskTokens then help access these capabilities without full fine-tuning.
>
> ## "What's the reason behind this method taking equivalent amount of environment steps to reach success as full-fine tuning?"
>  One likely reason is that we used the same hyperparameters as MaskedMimic, which may not be fully optimized for TaskTokens. We also observe that full fine-tuning tends to make faster initial progress, whereas TaskTokens (and PPO) show a delayed “jump” in performance later in training. This suggests that while TaskTokens is more parameter-efficient, its learning dynamics differ, and there may be room to accelerate early-stage learning with task-specific hyperparameter tuning.
>
> [1] Tessler et al. MaskedManipulator: Versatile Whole-Body Manipulation (2025)
>
> [2] Gal et al. An Image is Worth One Word: Personalizing Text-to-Image Generation using Textual Inversion (2022)

---

### Author Response · Authors · 2025-11-20
**Common Concerns Across Multiple Reviewers**

We thank the reviewers for their constructive feedback and are excited that the reviewers found our paper *"Elegant and Novel Solution to Important Problem"* (up9Q), *"The approach is sound, makes sense, and simple"* (MgVR), *"The paper is well‑motivated…leverages the transformer token interface without touching the backbone"* (zBs9), and *"The paper is clearly written and the method is well motivated"* (Vh4Q).

Here, we provide responses to common questions raised by the reviewers. In addition, we respond to each individual reviewer for their unique questions.

## Scalability and Multi-Task Learning (up9Q, zBs9)

Several reviewers expressed concerns about the scalability of TaskTokens, noting that training one encoder per task might limit multi-task, compositional, or continual learning. The current approach trains one separate Task Encoder for each new task, and our scalability claim emphasizes per-task efficiency: each encoder is compact (~200k parameters) relative to the full 25M-parameter BFM, making it feasible to train many diverse tasks. Once trained, multiple TaskTokens can be selectively and sequentially activated at inference according to the task, enabling flexible multi-task deployment without retraining the underlying model. While a single Task Token handles only one task, training multiple TaskTokens enables multi-task, compositional, and sequential usage: different encoders can be efficiently switched based on the current sub-goal.

Regarding the concern that our tasks are primarily reactive motor skills and may not demonstrate “multi-stage sequential logic,” we view TaskTokens as a modular component within a broader robotics pipeline. Long-horizon tasks are typically handled by decomposing them into short- or medium-horizon sub-goals using reasoning models such as VLMs - precisely the regime where TaskTokens excel. By encoding and reusing these sub-goals efficiently, TaskTokens provide a practical building block for scalable multi-stage behaviors without retraining the underlying BFM.

## Human-likeness vs. Task Performance Trade-off (Vh4Q, up9Q)
While PULSE often produces more human-like motion, this does not imply that TaskTokens yields unnatural behavior. In fact, results as well as the provided site with videos show that it produces much more human-like motions than the other baselines. Our results point to a tradeoff: PULSE achieves higher realism but requires significantly more trainable parameters and computational effort per task. In contrast, TaskTokens offers a far more efficient adaptation mechanism - several orders of magnitude smaller - while still producing coherent behaviors. In addition, PULSE is limited to single-modal prompting, whereas TaskTokens naturally supports multimodal task specifications.

---

### Meta-Review · Area_Chair_YwRa · 2026-01-11

**Summary:**

The paper received two accepts (8) from Reviewers up9Q and zBs9, and two borderline accepts (6) from Reviewers Vh4Q and MgVR.
All reviewers acknowledged the core contribution as sound, simple, and effective - a parameter-efficient method for adapting behavior foundation models to new tasks by training lightweight task encoders while keeping the BFM frozen.

**Reviewer Concerns:**

Concerns addressed

The authors effectively addressed several key concerns. On scalability, they clarified that multiple TaskTokens can be selectively activated at inference and integrated with VLM-based task decomposition for long-horizon behaviors. Regarding the human-likeness versus efficiency trade-off, they acknowledged the limitation transparently and noted that trust-region constraints could mitigate this in future work. The authors also provided clarification on task encoder architecture details (MLP with ReLU, running mean-std normalization) and explained the multi-phase prompting mechanism using a simple finite-state machine.

Outstanding concerns

Several concerns remain unresolved. No PEFT/LoRA baselines were provided, though reviewers acknowledged this is a reasonable scope limitation. The evaluation uses only one BFM (MaskedMimic), with manipulation deferred to future work. Some claims about parameter efficiency and robustness improvements require toning down in the final version. Additionally, the AMP baseline comparison methodology could be strengthened with task-specific motion data.

**Reviewer Scores:**

Reviewer up9Q (Initial: 8): Explicitly stated satisfaction with the authors' responses and confirmed maintaining their positive rating. No change expected.

Reviewer zBs9 (Initial: 8): Engaged substantively in discussion and acknowledged clarifications on scalability, training stability, and multi-phase prompting. While noting that the "order of magnitude" robustness claims remain overstated and that LoRA baselines would strengthen the work, this reviewer did not indicate intent to lower their score. Expected to maintain 8.

Reviewer MgVR (Initial: 6): Explicitly stated they are keeping their current score after the rebuttal. The reviewer's concerns about technical presentation of the task encoder and baseline inconsistencies were partially addressed but not fully resolved. No change expected.

Reviewer Vh4Q (Initial: 6): Did not respond post-rebuttal. Given that the authors addressed their questions about manipulation extensions, behavior elicitation, and the PULSE comparison with reasonable explanations, I would expect this reviewer to maintain or slightly increase their score. Projected to remain at 6.

The paper makes a clear, well-motivated contribution at the intersection of behavior foundation models and reinforcement learning. The Task Tokens approach is conceptually elegant - inspired by textual inversion in vision-language models - and demonstrates competitive performance with substantially fewer trainable parameters. The preservation of multi-modal prompting capabilities and out-of-distribution robustness are meaningful practical benefits.

The outstanding concerns are primarily about scope (single BFM, no manipulation) and missing comparisons (PEFT baselines) rather than fundamental methodological flaws. The core claims about parameter efficiency and preserved generalization are well-supported by the experiments. The average score of 7 with no reviewer below 6 indicates sufficient consensus for acceptance, and the paper would make a solid poster contribution to ICLR.

---

### Decision · Program_Chairs · 2026-01-26

Accept (Poster)